# Evaluation of Polyacrylonitrile Nonwoven Mats and Silver–Gold Bimetallic Nanoparticle-Decorated Nonwoven Mats for Potential Promotion of Wound Healing In Vitro and In Vivo and Bone Growth In Vitro

**DOI:** 10.3390/polym13040516

**Published:** 2021-02-09

**Authors:** Meng-Yi Bai, Fang-Yu Ku, Jia-Fwu Shyu, Tomohiro Hayashi, Chia-Chun Wu

**Affiliations:** 1Graduate Institute of Biomedical Engineering, National Taiwan University of Science and Technology, Taipei 10607, Taiwan; mybai@mail.ntust.edu.tw (M.-Y.B.); shyujeff@ndmctsgh.edu.tw (F.-Y.K.); 2Adjunct Appointment to the Department of Biomedical Engineering, National Defense Medical Center, Taipei 11490, Taiwan; 3Department of Biology and Anatomy, Tri-Service General Hospital, National Defense Medical Center, Taipei 11490, Taiwan; jiafwu.shyu@msa.hinet.net; 4Department of Materials Science and Engineering, School of Materials and Chemical Technology, Tokyo Institute of Technology, 4259 Nagatsuta-cho, Midori-ku, Yokohama 226-8503, Kanagawa, Japan; tomo@mac.titech.ac.jp; 5Department of Orthopaedic, Tri-Service General Hospital, National Defense Medical Center, Taipei 11490, Taiwan

**Keywords:** polyacrylonitrile, silver nanoparticles, electrospinning, Ag–Au bimetallic nonwoven mat, bone ingrowth, bone defect, soft tissue defect, urchin-like

## Abstract

We prepared polyacrylonitrile (PAN) and urchin-like Ag–Au bimetallic or Ag nanoparticle-decorated PAN nonwoven mats using electrospinning and evaluated them in vitro and in vivo for wound healing, antibacterial effects on skin tissue, and promotion of bone ingrowth in vitro. A facile, green, low-temperature protocol was developed to obtain these nonwoven mats. The sterilization rate of urchin-like Ag–Au bimetallic and Ag nanoparticle-decorated PAN nonwoven mats against *Staphylococcus aureus* was 96.81 ± 2.81% and 51.90 ± 9.07%, respectively, after 5 h treatment. In an in vitro cell model, these two mats did not show significant toxicity; cell viability of >80% was obtained within 5 h of treatment. In vivo animal model preclinical assessment showed that the urchin-like Ag–Au bimetallic nonwoven mat group showed significant wound recovery because of sebaceous gland, hair follicle, and fat formation during skin tissue regeneration; increased neovascularization and compact collagen fibers were observed in the dermal layer, comparable to the findings for the control group. The mother substrate of the urchin-like Ag–Au bimetallic nanoparticle-decorated PAN nonwoven mats, that is, pure PAN nonwoven mats, was found to be a potential scaffold for bone tissue engineering as osteoblast ingrowth from the top to the bottom of the membrane and proliferation inside the membrane were observed. The key genetic factor Cbfa1 was identified as a key osteoblast differentiation regulator in vitro. Thus, electrospun membrane materials show potential for use as dual-functional biomaterials for bone regeneration and infection control and composite grafts for infectious bone and soft tissue defects.

## 1. Introduction

Silver (Ag) is a powerful bactericidal substance that has been used since ancient times. Recent studies have revealed three plausible mechanisms underlying its antimicrobial effects: First, absorption of free Ag ions followed by disruption of ATP production and DNA replication. Second, Ag nanoparticles may release Ag^+^, thereby leading to the generation of reactive oxygen species (ROS) that attack microorganisms. Third, Ag nanoparticles directly damage the cell membrane. It is expected that the use of Ag-based antimicrobial materials would help overcome the hurdle of antibiotic resistance encountered with commonly used antibiotics, thereby making Ag the rising star of new antibacterial materials [1].

Various formulations containing Ag-based compounds have been used in many antimicrobial applications since the nineteenth century. Among them, Ag nanoparticles (AgNPs) have attracted the most attention owing to their unique physical, chemical, and biological properties. AgNP technology has been used in various processes and products [1]; AgNPs can be used in liquid form, such as gels, suspension coatings, and sprays, added to a solid matrix such as polymeric substrates, or suspended in a solid such as soap. They can also be incorporated into fiber spinning or filtration membranes for water-purification systems [2]. From the different types of solid matrix material available, electrospun polyacrylonitrile (PAN) has been widely investigated owing to its thermal stability, chemical resistance, and enhanced mechanical properties [3]. In addition, it has been widely used as a filter material, as well as for nanocomposites and other applications [4,5,6]. PAN is inert, strong, and stable, making it a suitable matrix material for biomaterial production for musculoskeletal systems, such as for promoting bone growth and wound healing [7].

In the musculoskeletal system, the most frequently noted effects of high-energy trauma (missile, vehicle traffic injury, crush or blast injury, falling from heights) are skin and soft tissue wound defects and even loss of some bony tissue. Considering the limitations of current therapies for bone defect management, bone tissue engineering (BTE) has drawn considerable attention in the past few decades [8,9,10,11,12]. Osseous defects caused by segmental bone resection due to tumor, trauma, or infection pose a great challenge for reconstruction and restoration of limb function. BTE focuses on the development of bone regeneration strategies involving loading of osteogenic cells onto various materials. One of the most common approaches is the development of a porous, bone structure–mimicking, three-dimensional (3D) scaffold with appropriate mechanical strength. Various materials such as metals, ceramics, polymers, and carbon-based nanomaterials are used to fabricate BTE scaffolds. Metal scaffolds for BTE applications, which provide superior mechanical properties, have been widely studied in recent years [13]. However, these scaffolds have several disadvantages, including stress shielding, osteointegration, and toxic metal ion and particle release [12]. Ceramic-based scaffolds are osteoinductive and biodegradable but are exceedingly brittle and have low mechanical strength [14]. Polymer-based scaffolds have great potential for use in BTE because of their superior biocompatibility, flexibility, and high biodegradability [14]. Musculoskeletal injuries often lead to skin and soft tissue wound defects, along with bone fractures. Therefore, while treating bone defects in the musculoskeletal structure, it is important to cover the wound next to the bone defect. With the advances made in biomedical materials, the medical dressings used to cover wounds now differ from traditional dressings such as cotton gauze.

The purpose of this study was to develop a new material with dual modalities for treating musculoskeletal trauma complicated with bone and soft tissue defects with the assistance of nanomaterials. The fine pores and urchin-like Ag–Au bimetallic nanoparticle-decorated block microbial invasion from the outside, and the fibrous PAN nonwoven mat substrate is expected to guide the migration of cell-like osteoblasts to the fracture to heal defects.

## 2. Materials and Methods 

### 2.1. Materials

PAN (M_w_ ≈ 150 kDa; 181315-50G; Sigma-Aldrich, St. Louis, MO, USA), Ag nitrate (S8157-10G, >99%, Sigma-Aldrich), l-ascorbic acid (A2174-100G, Sigma-Aldrich), dimethyl sulfoxide (DMSO; ≥99.7%, Sigma Aldrich, St. Louis, MO, USA), Dulbecco’s Modified Eagle Medium (DMEM; 21063-029; 500 mL; Gibco, Grand Island, NY, USA), and thiazol blue tetrazolium bromide (MTT; M2128-1g; 98%; Sigma Aldrich, St. Louis, MO, USA) were used without further purification. *N*,*N*-Dimethylformamide (reagent grade; Echo Chemical Co., Ltd., Miaoli, Taiwan) was used as the solvent in the electrospinning process. 

### 2.2. Preparation of Ag-Containing PAN Nonwoven Mat

PAN and AgNO_3_ were dissolved in DMF and stirred homogenously at room temperature to prepare the stock solution for the subsequent electrospinning process. We used a 1 mL syringe (inner diameter, 4.7 mm) to store the stock solution and attached a 20G needle (inner diameter, 0.6 mm; outer diameter, 0.9 mm; length, 25.4 mm) and then placed this apparatus in the injection pump (New Era Pump Systems, Inc., Syringe Pump NE-300, St. Farmingdale, NY, USA). A 10 × 10 cm^2^ iron sheet was utilized as an electrically grounded collector. The positive electrode of the high-voltage power supply was clamped on the needle, and the grounded wire was clamped to the iron sheet collector. The parameters were set as follows: voltage, 17 kV; flow rate, 7 μL/min; working distance, 12 cm to maintain a stable corn-jet mode electrospinning. After all the parameters were precisely set and a high-voltage power supply was applied, an electrical field was established between the capillary tube and collection substrate, and the electrospun PAN nanofibers were deposited on the collector. During the electrospinning process, we controlled the humidity of the working environment at approximately 40–50% for the pure PAN nonwoven mat (dehumidifier, Kolin Inc., Taipei, Taiwan) and approximately 20–30% for the AgPAN nonwoven mat by using a dehumidifier.

### 2.3. Generation of Ag NPs and Their Conversion to Urchin-Like Ag–Au Bimetallic NPs on PAN Nonwoven Mats by Using Galvanic Reaction and Calcined Ag/PAN Nonwoven Mats

Using ascorbic acid as the reducing agent, the Ag ions trapped in PAN were reduced to Ag NPs. Ascorbic acid (0.32 g) was added to deionized water (25 mL) and then the electrospun AgPAN dressing was soaked in the solution for 2 min to generate army green Ag NPs on an army green PAN nonwoven mat (Ag/PAN).

For the Ag–Au bimetallic NP-decorated PAN nonwoven mat, the abovementioned freshly prepared Ag/PAN was immersed in HAuCl_4 (aq)_ solution (0.25 g HAuCl_4_ dissolved in 25 mL deionized water) for the following galvanic reaction for 1 min: 3Ag^0^ + AuCl_4_ → Au + 3Ag^+^ + 4Cl^−^(1)

After the galvanic reaction, the nonwoven mat was placed in an oven (JUN YANG instrument co., LTD., New Taipei city, Taiwan) heated to 37 °C for 1–3 h for production of the urchin-like Ag–Au bimetallic NP-decorated PAN nonwoven mats.

For the production of calcined Ag/PAN nonwoven mats, we placed the AgPAN into the tubular high-temperature furnace (tube Furnace, Strong Youth Co., LTD., Taipei, Taiwan), the atmosphere inside was evacuated to 30 mmHg and filled with nitrogen. We set the heating procedures as follows: heating to 900 °C with rate 5 °C/min and the temperature was held at 900 °C for 1 h. After it was naturally cooled down to room temperature, the sample was taken out. The whole process was carried out at a rate of 50 mL/min under nitrogen gas flow.

### 2.4. Characterization

The morphological features of the nonwoven mats were observed using a high-resolution field-emission scanning electron microscope (FE-SEM; JSM-6500F; JEOL, Tokyo, Japan). The nanofiber diameters, Ag nanoparticle diameters, and nanofiber porosity were all determined by analyzing the SEM images with the image analysis software ImageJ (NIH, Bethesda, MD, USA). Raman spectra were measured with a commercial Raman microscope (Nanofinder 30, Tokyo Instruments Inc., Tokyo, Japan). The excitation wavelength was 532 nm. 

### 2.5. Measurement of In Vitro Ag-Ion Release 

The amount of Ag ions released from the Ag or urchin-like Ag–Au NP-decorated nonwoven mats was determined using inductively coupled plasma-optical emission spectrometry (ICP-OES; iCAP PRO; ThermoFisher Scientific, Waltham, MA, USA). A nonwoven mat piece (25 mg) was placed in a 15 mL vial, and 10 mL distilled water was added to the vial as the elution medium. The vial was shaken at approximately 100 rpm and incubated at 37 °C in a water bath. At the desired time point, all the medium was removed and replaced with 10 mL fresh medium to maintain a constant volume. Subsequently, all of the elution media collected were used for ICP-OES analyses, and the data for the amount of Ag ions released were used to establish a release profile.

### 2.6. Cytotoxicity of PAN Nonwoven Mats, Ag NPs, and Ag–Au Bimetallic Nanoparticle-Decorated Nonwoven Mats toward 3T3 Cells

To confirm whether Ag or Ag–Au-decorated nonwoven mats were toxic to skin cells, an MTT assay was performed using a mouse fibroblast cell line (NIH-3T3, Bioresource Collection and Research Center, Hsinchu, Taiwan) to test the cytotoxicity of various types of nonwoven mats toward cells in vitro. Basically, an extract solution method was adopted to conduct the MTT assay. Cells (n = 3, 5 × 10^4^/well) were seeded in 24-well plates (Biofil, Guangzhou Jet Bio-Filtration Co., Ltd., Guangzhou, China), and medium was added to obtain a total volume of 1 mL; subsequently, the plates were cultured in an incubator for 24 h (1st well plate). Then, 5 mg of pure PAN, Ag/PAN, and Ag–Au/PAN nonwoven mats was weighed and irradiated with UV light (254 nm) for 30 min sterilization. The same concentration of commercially available nano-Ag suspension kept steady for 24 h was used as the positive control, and medium without soaking of any samples was used as the negative control. The weighed nonwoven mats were immersed in 2 mL serum-free medium in another 24-well plate (2nd well plate) and placed in the incubator (Sanyo Electric Co., Ltd., Gifu, Japan) for 24 h. After 24 h of extraction, the medium in the 1st well plate was removed and the cells were washed twice with 1 mL phosphate-buffered saline (PBS, 10X, UniRegion Bio-tech, Miaoli, Taiwan). Subsequently, 500 μL of the nonwoven mat’s extract solution was taken from the 2nd well plate and added to the cell plate (1st well plate). After the predetermined treatment time, the medium was removed from the plate and MTT solution (thiazol blue tetrazolium bromide (MTT; M2128-1G; 98%; Sigma-Aldrich, 0.5 mg/mL) was added to each well to react with enzyme in the mitochondrial of the cell; the plate was subsequently incubated for 2 h. Then, MTT solution was replaced by DMSO and the plate was incubated for 10 min. Next, absorbance was measured at 570 nm using a UV-Visible spectrophotometer (BMG Labtech, Ortenberg, Germany).

### 2.7. Evaluation of Antibacterial Effect 

The antibacterial effect of various types of nonwoven mats was evaluated using two methods. The first method was the disc diffusion method [8]. The inhibition zone was analyzed using the qualitative disc diffusion method with 10^9^ colony-forming units (CFU) of *Staphylococcus aureus* cultured on LB agar plates and placed on 8 mm nonwoven mat samples. The cultured plates were then incubated for 24 h at 37 °C in an incubator, following which the inhibition zone of the microbial colony was observed and measured (OLYMPUS CORPORATION, Tokyo, Japan).

The second method involved quantitative evaluation in a bacterial culture solution test [9]. The nonwoven mats were cocultured with *S. aureus* LB broth suspension. The bacterial solution was sampled from the incubated LB broth at predetermined time points (1, 5, and 24 h) and immediately streaked on nutrient agar (Gibco LB Broth, Liquid, Thermo Fisher Scientific, MA, USA) plates using four-way streak plate inoculation. The plates were then incubated at 37 °C for 16–20 h, and the colonies on the plate were counted for each sample. A plot of colony-forming units (CFU/mL) of *S. aureus* versus time was prepared for quantitative analysis of the antibacterial rates for each sample.

### 2.8. Skin Sensitization Study

Tegaderm^TM^ film (6 × 7 cm^2^; 3M Healthcare, Neuss, Germany) was used as the backsheet for immobilizing the nonwoven mat. Pure PAN, Ag/PAN, and Ag–Au/PAN nonwoven mats (size, 1 × 1 cm^2^) were cut and stuck to the middle of the Tegaderm^TM^ film. This animal test is consistent with and approved as per the animal use protocol of the Institutional Animal Care and Use Committee (IACUC) of the National Defense Medical Center (NDMC; certificate no. IACUC-17-305). All procedures were in accordance with the ISO 10993-10:2002 tests for irritation and skin sensitization specifications. Our sensitization and irritation tests were thus based on ISO10993-10 guidelines, which suggest suitable animal species and time-period 3 days and 1–3 weeks to see the acute and long-term response of the sensitization and irritation evaluation. To study allergic reactions to the nonwoven mat, guinea pigs with high skin sensitivity were utilized. The dorsal surface of the guinea pigs was shaved before the test, and a 2 × 4 cm^2^ shaved area was used as the test area for clear observation. In this preclinical animal test, the positive and negative control groups included animals for whom a skin irritant chemical and pure PAN, respectively, had been used. The experimental groups were Ag/PAN and urchin-like Ag–Au/PAN. The test substances for each group were then applied to the test area and covered with 3M dressing (Tegaderm™, St. Paul, MN, USA). We continuously monitored skin sensitivity every day for 7 days long.

### 2.9. Skin Irritation Study

The skin irritation study was similar to the skin sensitization study, except for the animal species used. A 3M Tegaderm™ film (6 × 7 cm^2^) was used as a base dressing for adherence to the nonwoven mats. Pure PAN, Ag/PAN, and urchin-like Ag–Au/PAN dressings of appropriate sizes were cut and stuck to the middle of the 3M dressing. The procedures used adhere to the ISO 10993-10:2010 guidelines for irritation and skin sensitization specifications. We applied the test substances directly on the skin surface of the animal for daily observation. In this experiment, New Zealand white rabbits were used as test subjects.

The rabbits were shaved before the test, and a 2 × 4 cm^2^ area was marked as the targeted test area in the shaved region. The positive and negative control groups included animals for whom a skin irritant chemical and pure PAN, respectively, had been used. The experimental groups were PAN, Ag/PAN, and urchin-like Ag–Au/PAN (n = 3/group). The test dressings of each group were covered in the test area and covered with 3M dressing (Tegaderm™). Skin conditions were observed and recorded after 24, 48, and 72 h.

### 2.10. In Vivo Evaluation of the Wounded Mouse Model 

C57BL/6JNarl mice were used for this analysis due to their thicknesses of full skin are similar to that of human, and their sizes are convenient for animal fixation to conduct full skin removal surgery; the experiment was performed on the dorsal surface of the mice (n = 5/group). Their backs were shaved, and they were anesthetized with Zoletil^TM^ (20–40 mg/kg, Virbac, Carros, France). A 1 × 1 cm^2^ wound was created on the back of the mice by performing a full skin removal surgery. The previously cultured *S. aureus* bacterial suspension was diluted to 10^8^ CFU/mL in PBS. Then, 150 μL bacterial suspension was dropped onto the wound by using a micropipette. After 1–3 min, we wrapped the wound with 3M dressings and maintained wound induction in the infected wound model for 1 week. After 7 days of wound infection, the pus on the wound was collected and suspended in 1 mL PBS. The mixture was vibrated for 50 s with an ultrasonic processor and centrifuged at 1000 rpm for 8 min at 4 °C. The supernatant was cultured for 20 h before performing the plate count. After the wound was cleaned and the remaining pus removed, the wound was wrapped with various types of dressings, and the dressing was replaced every 6 days or 2 days for Ag/PAN nonwoven mat and urchin-like Ag–Au/PAN nonwoven mats, respectively, as these two types of dressing show different ability on antibacterial and extrudate absorption ability.

### 2.11. Pathological Analysis

The healed mice were euthanized, and the regenerated skin area of the dorsal wounds and the liver of the mice were harvested. After the specimens were maintained in formalin for 24 h, they were sent to the Taipei Pathology Center for pathological examination, and Masson, hematoxylin and eosin (H&E) staining was performed. The pathological sections were analyzed using microscopy, and the newborn skin and original skin tissues were compared.

### 2.12. Culture and Functional Assay of Osteoblasts/Immunofluorescence Staining

The osteoblasts were isolated from 1-day-old newborn rats as previously described [15]. Briefly, osteoblasts were harvested from calvariae and cultured in α-MEM medium with 10% FBS, 0.05% ascorbic acid (sc-394304, Santa Cruz Biotechnology, Santa Cruz, CA, USA), 2 mM β-glycerophosphate (G9422, Cayman Chemical, Ann Arbor, MI, USA), 10 nM dexamethasone (sc-29059B, Santa Cruz Biotechnology). Medium were changed every 3 days. Osteoblasts of 10^5^/cm^2^ were seeded into a 6-well plate containing PAN with the same medium for 28 days. Then, osteoblasts were fixed and labeled with primary anti-Core-binding factor alpha1 (Cbfa1, ab23981, Abcam, Cambridge, MA, USA) antibody and FITC-conjugated secondary antibody. Actin was labeled by iFluor™ 546-Phalloidin (22663, AAT Bioquest, Sunnyvale, CA, USA). Cell nuclei were visualized by DAPI staining. Imagines of the fluorescently stain osteoblast cells were taken by using confocal microscope (LSM 880, Zeiss, Oberkochen, Germany). 

### 2.13. Statistical Analyses

Data are expressed as mean ± standard deviation (SD). Statistical analyses were performed using SPSS software (SPSS, Chicago, IL, USA). Specifically, Wilcoxon statistics, one-way analysis of variance, and the Student’s *t* test were used to assess the differences between the experimental groups. The data were considered significantly different when *p* < 0.05.

## 3. Results and Discussion 

### 3.1. Characterization of Surface Morphological Features of Ag/PAN and Urchin-Like Ag–Au/PAN Nonwoven Mats

Figure 1 shows the surface morphological features of the nonwoven mats, as observed using FE-SEM. The nanofibers were bent and partly twisted and were randomly entangled with each other. The fiber diameters of pure PAN, Ag/PAN, Ag/calcined PAN, and Ag–Au/PAN were 440.4 ± 123 nm (N = 50), and fell within 47.20%, 38.03%, and 32.29% reduction in diameter in other three samples, respectively. This significant decrease could be attributed to the prominent promotion of conductivity of the stock solution noted after the Ag and Au precursors were added. In addition, the wound extrudate could be adsorbed to prevent infiltration and maintain moisture in the wound, which is an ideal feature for dressing applications. The average surface area for the samples was 13.1 m^2^/g and 11 m^2^/g. The slight reduction in particle size in the Ag–Au/PAN group was due to the decoration of urchin-like Ag–Au particles with big size on the fiber surface; the size of the Ag–Au particles was much larger (1.1 ± 0.28 μm) than that of Ag nanoparticles (162.5 ± 23.9 nm).

### 3.2. Raman Spectrum Analysis

Figure 2 shows a comparison of the peak intensities of the calcined pure PAN and Ag/PAN nonwoven mats subjected to high-temperature (900 °C) treatment, which demonstrates that the PAN nonwoven mats could be further converted to the useful graphite-like structure at relatively low thermal treatment temperature. The chemical structure of the PAN nonwoven mat was cyclized owing to the high-temperature ring opening and cyclization process. Both G (graphite-like structure) and D (disordered carbon structure) peaks appeared in the Raman spectra of both samples, but the peak intensities of the two samples were significantly different. On comparing the G peaks for the intensities of pure PAN and calcined Ag/PAN were found the strength of the intensity was enhanced by approximately six times because of the surface-enhanced Raman scattering (SERS) effect of the Ag nanoparticles. This phenomenon confirms that the SERS effect of the Ag NPs decorated on the PAN nonwoven mats shows potential to enhance the adsorption of molecules on the surface of rough metal and nanostructures for wound infection monitoring and detection [16].

### 3.3. In Vitro Measurement of Ag-Ion Release Profile 

In the case of Ag/PAN, a relatively low concentration of Ag ions was released within 5 h (avg., 0.062 ± 0.03 ppm) (Figure 3). In contrast, the Ag–Au/PAN nonwoven mats showed an initial burst release of Ag ions with concentrations up to an average of 1.81 ± 0.23 ppm being obtained, possibly due to galvanic corrosion in the presence of an electrolyte to facilitate the release [17]. The Au ions were synergistically released with Ag ions and their concentration abruptly increased to 0.4 ± 0.1 ppm in the subsequent lag phase with Ag ions (5.7 ± 1.0 ppm). This phenomenon may be due to galvanic corrosion when the bimetallic Ag–Au NPs contacted the electrolyte in the PBS solution.

### 3.4. Cell Toxicity

Figure 4A shows the results of the cytotoxicity assays with 3T3 fibroblasts using Ag/PAN or calcined Ag/ PAN, respectively. The negative control group was used as the reference standard, and the results for all the other groups were normalized to those for the control group. Based on the ISO10993 guideline, 80% (as red line indicated in Figure 4) of cell viability was set us the threshold to define whether significant cytotoxicity was observed. After treatment, the cell viability for the groups was as follows: pure PAN, 97.8 ± 3.0%; Ag/PAN, 80.2 ± 1.1%; diluted solution of calcined Ag/PAN, 87.7 ± 0.3% (this 40% diluted concentration was equivalent to the concentration of Ag/PAN released for 24 h, the maximum time applied when used in animal model). The cell viability of all the other groups except 100% and 80% diluted concentration of extract silver solution were above 80%, which indicated that none of the materials showed significant toxicity when the concentration of silver ion below 1.22 ppm. In addition, urchin-like Ag–Au/PAN nonwoven mats were subjected to cytotoxicity assays independently, because their release profiles and release components differed from those of Ag/PAN or calcined Ag/PAN nonwoven mats. Figure 4B shows the cytotoxicity assay results for 3T3 fibroblasts after treatment with an extract solution of urchin-like Ag–Au/PAN nonwoven mats. The Ag/PAN and urchin-like Ag–Au/PAN nonwoven mats showed significantly different release profiles and cytotoxicity toward 3T3 fibroblasts. The cell viability of Ag/PAN-treated groups was above 80%. In contrast, in the urchin-like Ag–Au/PAN nonwoven mats treated group, the cell viability significantly decreased at all three time points of treatment (1, 3, and 5 h). The most plausible reason for this significant reduction in cell viability as compared to that of Ag/PAN nonwoven mats could be the galvanic corrosion that occurred when the bimetallic Ag–Au NPs contacted the electrolyte in the PBS solution, which promptly increased the concentration of the Ag^+^ released up to 0.553 ± 0.09 ppm within just 1 h of contact with the nonwoven mat to generate the acute toxicity. In general, in conjunction with the silver ion release profile shown in Figure 3 and in comparison to our unpublished data, we found that when the concentration of Ag is approximately above 5–6 ppm, significant cytotoxicity was observed. This why we can see that the cell viability was decreased when the Ag concentration is starting to increase as time passes by from 1, 3, and 5 h.

This finding is similar to that of an in vivo study by Maneewattanapinyo et al. wherein a small shaved area of skin on guinea pigs was exposed to 50 ppm AgNP (10–20 nm) for 24 h, and the animals were observed for signs of acute toxicity [18]. To strike a balance between acute cytotoxicity, antibacterial performance, and costs of using this nonwoven mat for potential applications such as dressings, the size of the material used can be decreased to reduce production cost (since noble metals like Ag and Au are being used) and possible cytotoxicity. 

### 3.5. Antibacterial Effects 

Figure 5A shows a schematic illustration of the inhibition zone test, and Figure 5B–D shows the findings for the qualitative inhibition zone test for Ag/PAN and calcined Ag/PAN nonwoven mats and the quantitative antibacterial assay based on the streak plate method. For the nonwoven mats without Ag nanoparticle encapsulation, no inhibition zones were observed around these specimens. For the Ag-containing nonwoven mats, the diameters of calcined Ag/PAN and Ag/PAN were 0.36 mm and 0.11 mm, respectively. Urchin-like Ag–Au/PAN nonwoven mats showed a significantly enhanced antibacterial effect toward *S. aureus* (Figure 5E diameter, 2.74 ± 0.33 mm and Figure 5F, 0 count of CFU after 5 h of treatment at avg., 0.062 ± 0.03 ppm of Ag ion concentration) because of their improved porosity, Ag ion release efficiency, and synergistical effect of Ag–Au co-release. This effective dosage of silver ion was relatively low as compared to another research works (Doaa Safwat Mohamed et al. reported 6–12 ppm to reach the same colony reduction toward *S. aureus*) [19]. This Ag–Au synergistical effect generates remarkable reduction in the Ag ion needed for showing antibacterial performance, which also reduces the cytotoxicity toward normal cells. Additionally, they exhibited excellent antibacterial effects against the Gram-negative *Escherichia coli*. In comparison to Ag/calcined PAN and Ag/PAN nonwoven mats (sterilization rate: 99% after 5 h treatment), urchin-like Ag–Au/PAN nonwoven mats were found to be a more effective and powerful material for antibacterial purposes with showing faster efficacy observed.

### 3.6. Skin Sensitization Study

The skin condition was defined according to the Magnusson and Kligman scale (ISO 10993-10; Appendix A) [20]. After the predetermined allergic induction test period, the findings for skin covered with various types of nonwoven mats were recorded (Figure 6). The positive control (CDNB group) showed obvious erythema and edema, while the negative control, Ag/PAN group, calcined Ag/PAN group and urchin-like Ag–Au/PAN nonwoven mats did not show any significant allergic reaction. Thus, the findings confirmed that the nonwoven mats did not cause significant allergic reactions.

### 3.7. Skin Irritation Study

The skin reaction scores in the irritation test after 24, 48, and 72 h of contact with different types of nonwoven mats were also evaluated, and the erythema and edema scores were evaluated on the basis of the Score System of Skin Reaction (Appendix A) [21,22]; the degree of irritability was also assessed (Figure 7). Analysis of the skin condition of the rabbits with the dressing showed that the positive control group had obvious erythema and edema, while the negative control, Ag/PAN, and calcined Ag/PAN nonwoven mat groups did not have erythema or edema. The experiment confirmed that the dressings used did not cause skin irritation.

### 3.8. In Vivo Evaluation of Infected Wound Mouse Model

We observed changes in the number of colonies after inoculation represent the quantification of infection. The initially inoculated colony count was 10^8^ CFU/mL. However, after 7 days of infection, the number of colonies of wounded skin in each group was as follows: Aquacel, 8.7 × 10^7^ CFU/mL; pure PAN, 4.3 × 10^8^ CFU/mL; Ag/PAN, 2.8 × 10^8^ CFU/mL; calcined Ag/PAN, 1.6 × 10^8^ CFU/mL; urchin-like Ag–Au/PAN nonwoven mats, no observable colonies. The commercially available Aquacel dressing had the best antibacterial effect, inhibited bacterial growth, and reduced the number of colonies as compared to those obtained for the only Ag-containing group. However, the bimetallic Ag–Au/PAN inhibited bacterial growth to a greater extent than the only Ag-containing group did, and no pus was observed after inoculation (please see Figure 8, right scale at day 0). Because pure PAN does not contain Ag, the bacteria grew faster and the inhibition effect was lower than that obtained with the other groups. Figure 8 shows images of the back wounds of mice, recorded daily after surgery, and infection (Day-7) until the end of wound closure. The rate of wound closure in mice is an important indicator for evaluating dressing effectiveness. Using the ImageJ software, the percentage of wound healing over time according to the initial surgical wound size can be calculated from the photograph. The statistical analysis results for the percentage recovery of wound area over time for all the mice (n = 5 for each group) show that when the Aquacel, pure PAN, Ag/PAN, calcined Ag/PAN, and urchin-like Ag–Au/PAN treated groups were restored to 50% of the initial area, the time spent was 1, 12, 29, 7, and 5 days, respectively. Additionally, pathological analysis shows that urchin-like Ag–Au bimetallic nonwoven mat group showed significant good wound recovery because of sebaceous gland, hair follicle, and fat formation during skin tissue regeneration; increased neovascularization and compact collagen fibers were observed in the dermal layer, comparable to the findings for the control group (see Appendix A for comparison).

Some studies reported that carbonized PAN (cPAN) has drawn attention in the BTE field in recent years. Compared to other carbon-based biomaterials, such as graphene and its derivatives, carbon nanotubes (CNTs), carbon dots, and cPAN can be easily manufactured as 3D scaffolds [23]. In an earlier study, Ryu et al. also prepared a 3D electrospun cPAN scaffold with controllable pore size; the scaffold prepared showed excellent biocompatibility and better osteoinductivity than CNTs because of enhanced osteogenic differentiation of mesenchymal stem cells [24]. In addition, cPAN has been used to manufacture a bone-mimicking 3D scaffold using the cryogel technique in another study; the scaffold prepared showed improved osteoconductivity because the adhesion and proliferation of MG-63 osteoblast-like cells were better than those obtained with cell culture–treated polystyrene well plates [25]. To enhance mechanical properties, Wu et al. fabricated 3D cPAN/hydroxyapatite (HA) composite scaffolds by growing HA crystals on carbonized electrospun PAN scaffolds. The fracture strength of the composite scaffolds was highly improved, making it a potential artificial bone graft material [26]. The abovementioned studies show that PAN-based materials might represent a potential candidate for bone-integrating orthopedic implants. 

Currently, poor formation of bounding at the interface between bone implant and infection represents a challenge in maintaining implant survival [27]. Modifying the biological response improves biocompatibility and integration with adjacent bone tissue on the orthopedic implant surface. Surface chemical properties play an important role when choosing biological osteointegration at the prosthesis–bone interface. It is becoming increasingly important to generate a microenvironment that stimulates osteointegration and improves the antibacterial effect of the surface used for fabricating implants. To strike a balance between novelty, mechanical strength, ease of stretching, and nature of osteoconductivity, PAN nonwoven mats were first used as a 3D scaffold for in vitro osteoblast culture to prove this novel idea and, thus, turn this preclinical fail antibacterial material into a new era. In the experiment showed in Figure 9, osteoblasts were seeded into a PAN nonwoven mat and cultured for 28 days. Images of confocal microscopy showed a group of osteoblasts covered on the surface as well as grew in 100 mm deep of PAN. High expression of the Cbfa1 (green staining), one of the major osteoblast transcription factors, indicates increase activity in these cells. Strongly labeling of stress fibers-like cytoskeleton (actin, red staining) indicates cell adhesion and osteoconductivity of PAN. Together, these results suggest osteoblasts survive and may proliferate in PAN. This proof-of-concept result showed that the PAN nonwoven mat is a potential biomaterial for use as a bone graft material, which strikes a balance between biocompatibility, mechanical properties, osteoblast biology response, and osteoblast induction.

## 4. Conclusions

In this study, a series of composite electrospinning Ag-containing PAN, converted carbon nonwoven mats, and urchin-like Ag–Au/PAN nonwoven mats were successfully prepared using an electrospinning system with PAN as a substrate, loaded with/or without Ag or Ag–Au nanoparticles. Several advantages were obtained using this approach: porosity was enhanced for guiding cells during growth, high sterilization efficiency was obtained against foreign invasion, no significant allergy or irritation was observed, and the dressing did not adhere to the skin, which is expected to aid in wound healing and union of bone fractures. On the basis of these findings, we anticipate the development of a double-sided prosthesis for dual modality use, such as the treatment of open segmental fractures with soft tissue defects in the future.

## Figures and Tables

**Figure 1 polymers-13-00516-f001:**
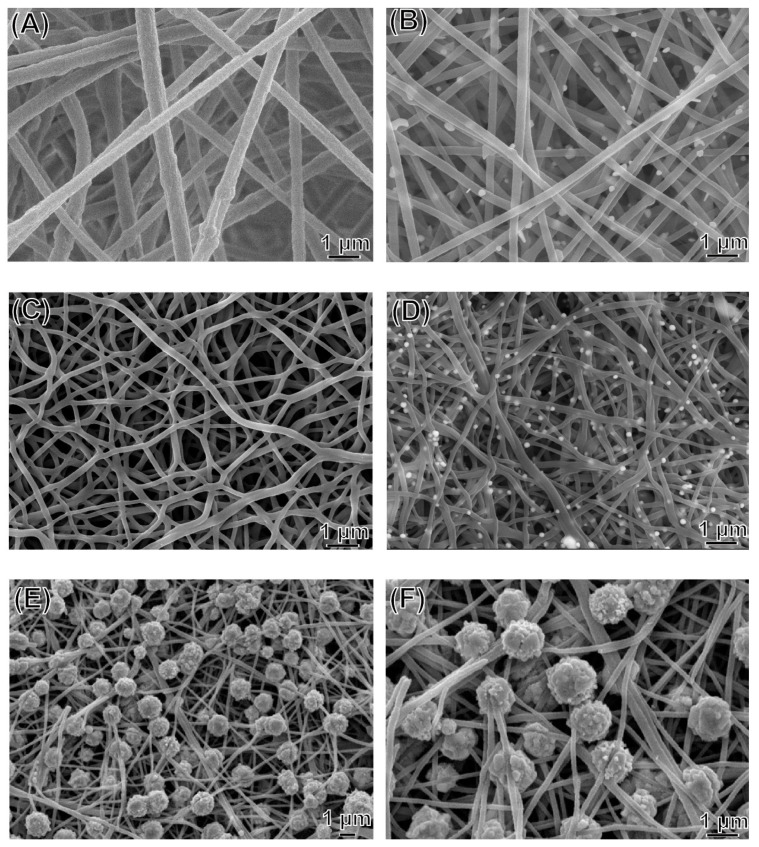
FE-SEM images of (**A**) pure polyacrylonitrile (PAN) nonwoven mats; (**B**) Ag/PAN nonwoven mats; (**C**) calcined pure PAN nonwoven mats; (**D**) calcined Ag/PAN nonwoven mats; (**E**,**F**) urchin-like Ag–Au/PAN nonwoven mats.

**Figure 2 polymers-13-00516-f002:**
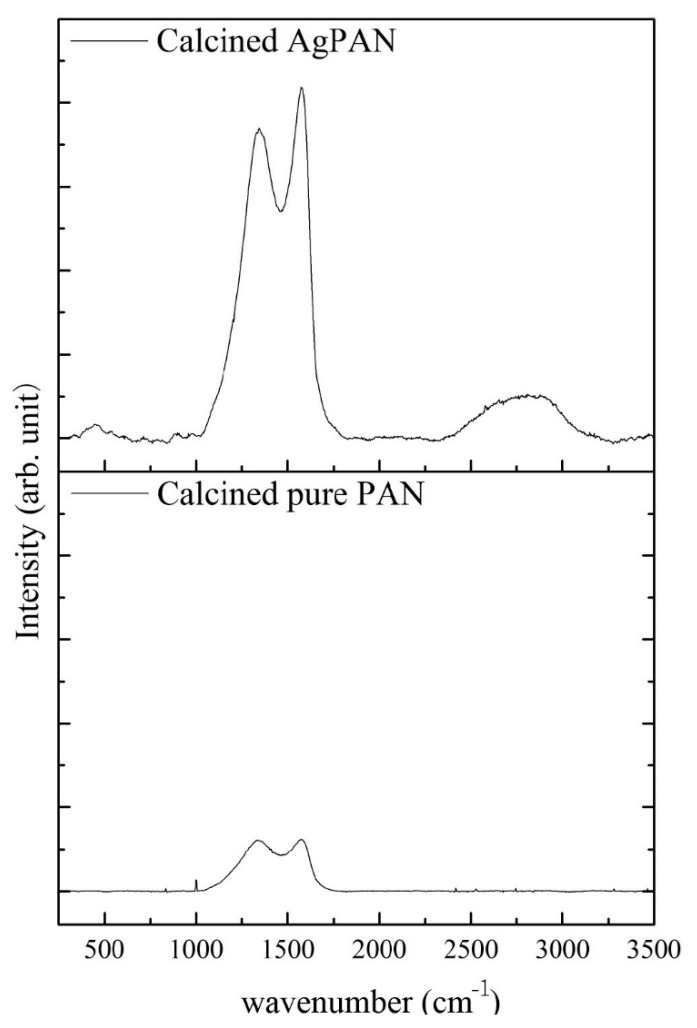
Raman spectrum of calcined pure PAN and Ag/calcined PAN.

**Figure 3 polymers-13-00516-f003:**
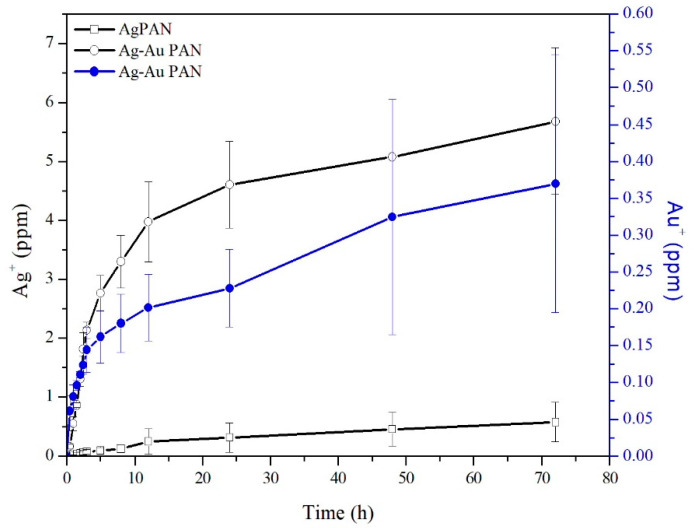
Profile of silver ion release curve.

**Figure 4 polymers-13-00516-f004:**
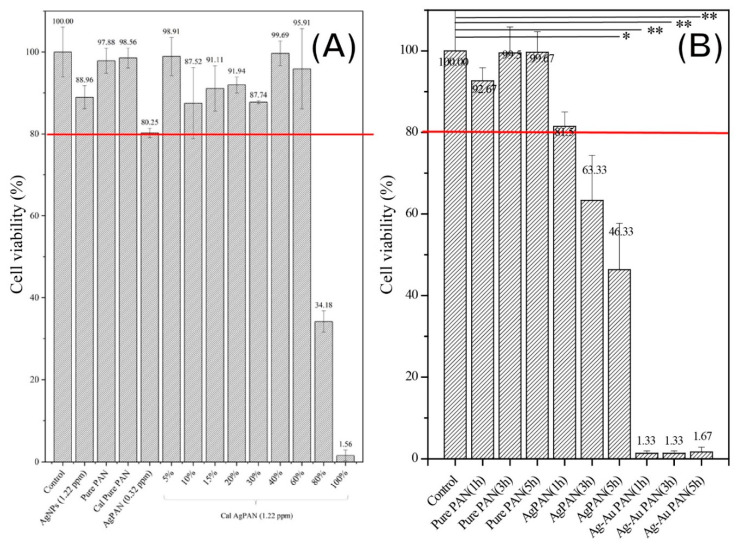
MTT assay results for the nanofibers. The results were analyzed using the *t*-test, with a significance level of *p* < 0.05. ** p* < 0.05; *** p* < 0.01.

**Figure 5 polymers-13-00516-f005:**
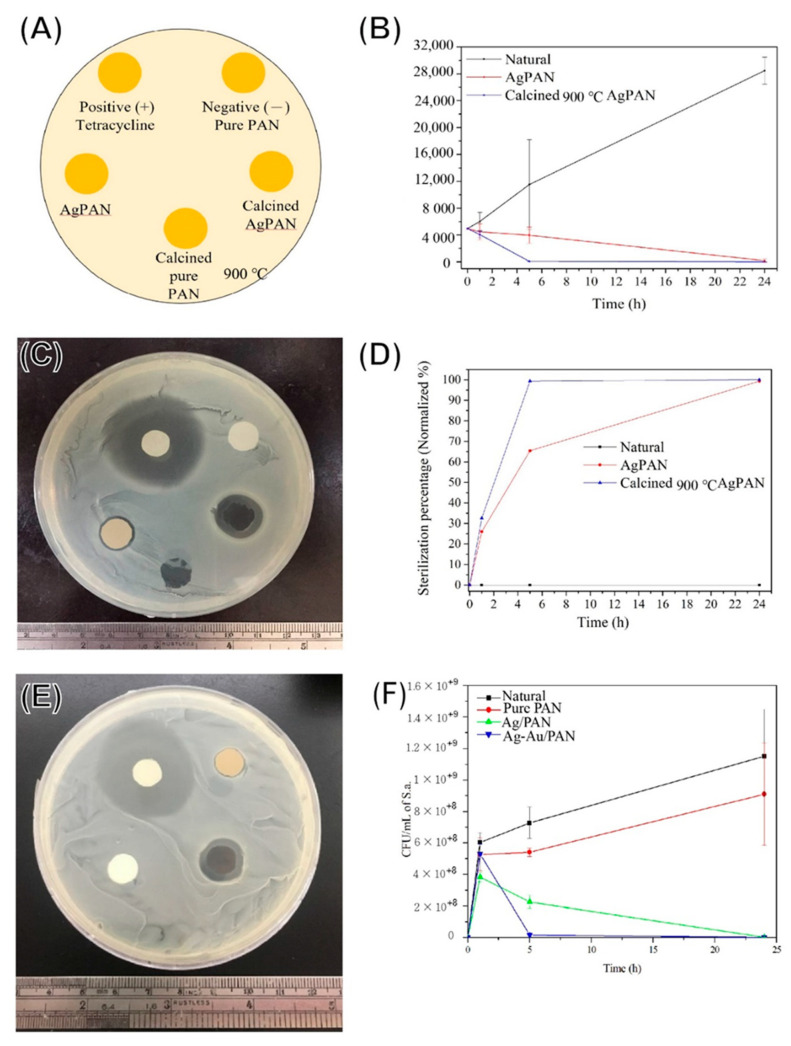
Antibacterial effect of PAN composites on the basis of the inhibition zone test and sterilization rate test. (**A**) Sample schematic illustration; (**B**) bacterial colonies numbers after being treated of different Ag/PAN composites; (**C**) qualitatively inhibition zone results of calcined Ag/PAN; (**D**) quantitively sterilization rate of (**B**) after being normalized to control group; (**E**) qualitative inhibition zone results of urchin-like Ag–Au/PAN nonwoven mats and (**F**) quantitive sterilization rate of urchin-like Ag–Au/PAN nonwoven mats.

**Figure 6 polymers-13-00516-f006:**
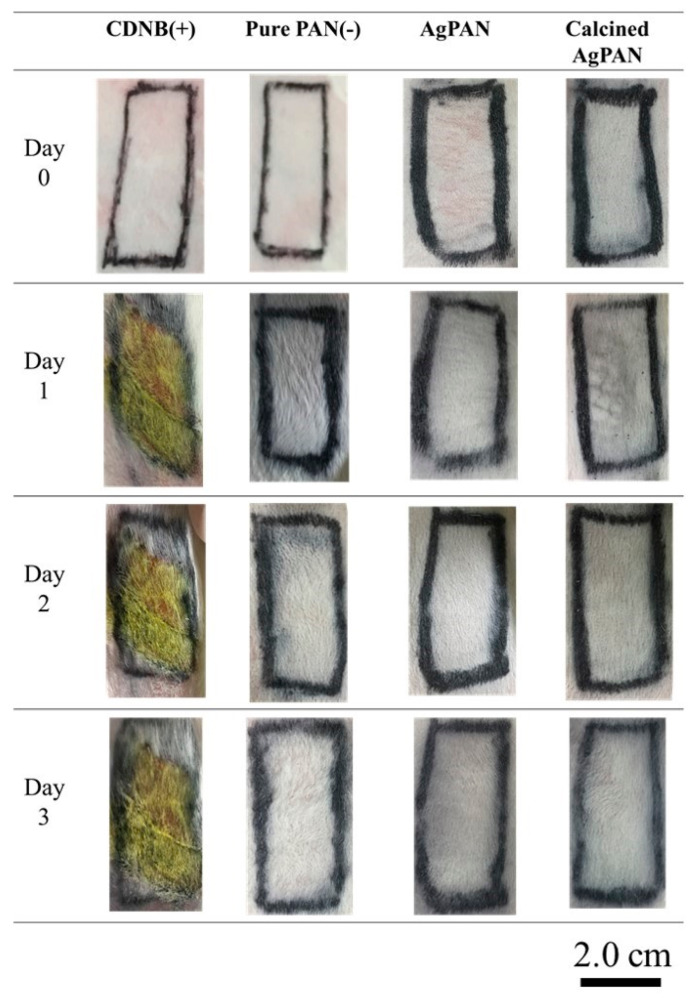
Testing for sensitivity of animal skin to various types of nonwoven mats for 3 days.

**Figure 7 polymers-13-00516-f007:**
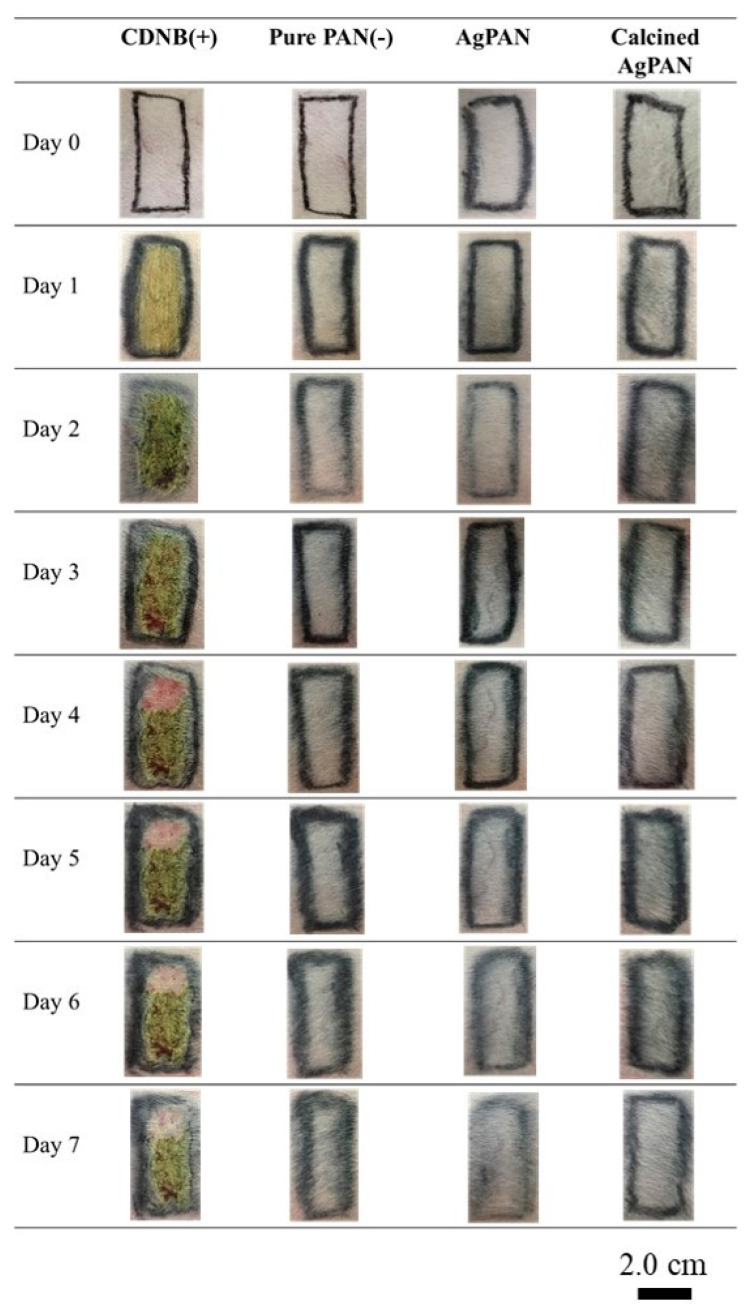
Testing for irritation of animal skin by various types of nonwoven mats for 7 days.

**Figure 8 polymers-13-00516-f008:**
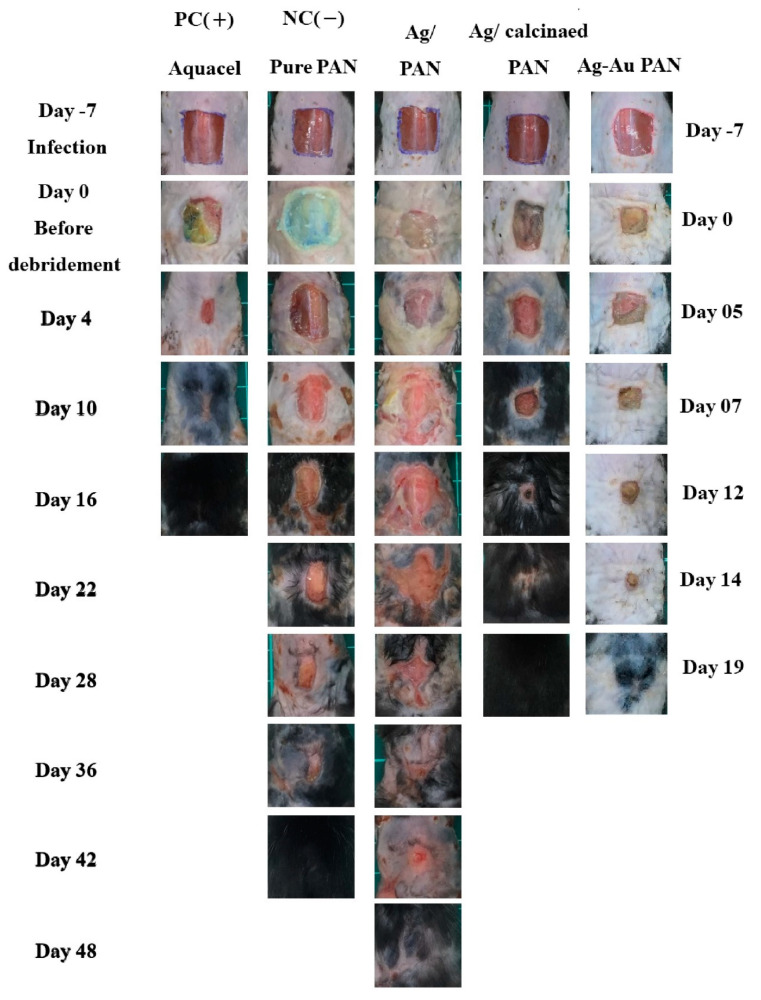
Optical images of two runs of animal studies for evaluating two major different nonwoven mats, i.e., Ag-based/PAN and urchin-like Ag–Au/PAN nonwoven mats. The left scale bar was versus group PC, NC, Ag/PAN, and calcined Ag/PAN nonwoven mats. The right scale bar was versus urchin-like Ag–Au/PAN nonwoven mats. Changes in wound size in mice in infection experiments and wound area recorded over time.

**Figure 9 polymers-13-00516-f009:**
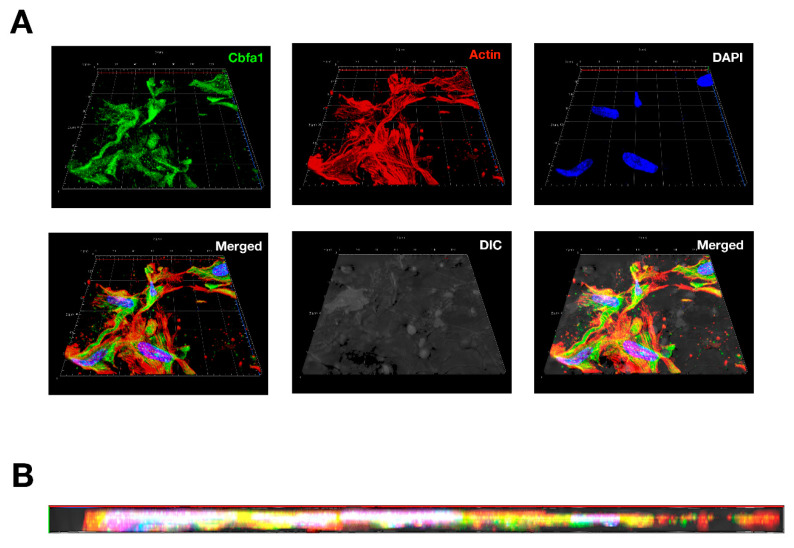
Confocal imagines of osteoblasts cultured on PAN nonwoven mats: (**A**) Top view of 3D confocal image of osteoblast seeding into specimen. The 130 × 130 mm confocal image were constructed from 10 consecutive 10 mm-thick sections. (**B**) Cross-section view of the constructed image as shown in (**A**). DAPI (blue), Cbfa1 (green), and actin (red) represent the staining method for labelling the nucleus, osteoblast-specific Cbfa1, and cytoskeleton of osteoblasts, respectively.

## Data Availability

The data presented in this study are available on request from the corresponding author.

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
