# Peer review of "Evaluation of Polyacrylonitrile Nonwoven Mats and Silver–Gold Bimetallic Nanoparticle-Decorated Nonwoven Mats for Potential Promotion of Wound Healing In Vitro and In Vivo and Bone Growth In Vitro"

_polymers, 2021, doi:10.3390/polym13040516_

Round 1
Reviewer 1 Report
The authors present PAN nonwoven mats decorated by Ag or Ag-Au. However they try to present complex view of the in vitro and in vivo characteristics of the mats. There are lot of serious inaccuracy in the figures, in the description of material and methods and in the interpretation of the results, which has to be clarified before future processing of the manuscript.
Comments:
- In generally, the English writing has to be checked. In a lot of description the authors used un-standard expressions. That is why the writing does not describe precisely the data, situation or ideas. g. …is an important engineering polymer material - why important?; …high mechanical properties…. What authors mean?;` …..high-energy trauma are skin and soft tissue wound defects and even loss of some bony tissue….. – what is high energy trauma?
- The authors did not perform any experiment expected test with growing osteoblast to clarify if the mats are sufficient for bon regeneration. It has to be done more precisely or the authors have to change the „current of the manuscript“ and avoid the hypothesis of bone regeneration without substantiation of the experiments.
- Which type of MTT assay was used? How many experiments were done and how many repetitions?
- In generally, the authors have to complete the methods with the information about number of animals, number of repetition of experiments. There is missing the description of which type of osteoblast was used, and cultivation conditions. There is missing the description of statistical analysis, software used and type of test in all experiments.
- It is not clear, thy author divide sensitization and irritation experiments and why used two types of animals and for wound healing the third type of animal? The description of the in vivo experiment in methods are different than description of time points on the figure. It has to be change and clarify.
- In the methods the authors write about the urchine like nonwoven mats, and calcined nonwoven mats are not describe. Through the results the authors sometimes analyzed calcined Ag/PAN, sometimes urchine. It is not cleare which type of the mats were analyzed in which experiment or if authors named the mats differently. It has to be clarified.
Preparation of calcined Ag/PAN nonwoven mats
- There is missing discussion in part of the release study, if the release of Ag, Au is sufficient in comparison with other studies.
- 4 the clear description of figure is missing, It is not clear what is A and B. There is presented different type of graphs, why? There is visible significant toxicity, without proper discussion in the text. It is completely unclear the descriptions on line X.
- In the Fig.5. simillar as Fig. 4 is missing clear description of the figure.
- In the text, where authors describe data from Fig. 6 is written different type of mats, than is presented on the Fig. 6. Moreover the authors presented only 3 days of experiment and on the Fig. 7 the 7 days of experiment. Why?
- The authors presented the healing potency of prepared mats, however on left site is different numbering of days than on the right side. There is not clear which days belong to which picture?
- The authors presented on Fig.9, the data which is not describe in the methods. There is missing the ruler on the Picture and there is missing the discussion about spreading, morphology and viability of the cells. On the other hand the authors describe in methods the pathology analysis, which is not describe in results and discussion.
Author Response
Response to Reviewer1 Comments
Dear editor and reviewers,
Many thanks for your e-mail dated 12/29/2020. We appreciate the review of our manuscript entitled “Evaluation of Polyacrylonitrile Nonwoven Mats and Silver–Gold Bimetallic Nanoparticle–Decorated Nonwoven Mats for Potential Promotion of Wound Healing In Vitro and In Vivo and Bone Growth In Vitro”. We are grateful for your generous and helpful comments on the manuscript. The manuscript has been revised and modified according to the reviewers’ concerns point-by-point below. We have made all the changes that were suggested by the reviewers. We would like to address your comments here:
For your information at the beginning of review process:
- Newly added paragraphs in the revised manuscript in response to the reviewers’ comments are highlighted in yellow color.
- For those typo errors indicated by reviewers’ are highlighted in red.
- For repetition or animal use numbers are highlighted in blue.
Comments from the reviewer
Reviewer # 1
- “In generally, the English writing has to be checked. In a lot of description the authors used un-standard expressions. That is why the writing does not describe precisely the data, situation or ideas. g. …is an important engineering polymer material - why important?; …high mechanical properties…. What authors mean?;` …..high-energy trauma are skin and soft tissue wound defects and even loss of some bony tissue….. – what is high energy trauma”
Response to Reviewer comment No. 1:
We thank this reviewer’s constructive comments. We have made the manuscript revised by an English native speaker. Please see certificate of revision attached as follows. Also, we have responded to all issues raised by the reviewer (please see those sentences highlighted with color yellow in abstract page 2, line31&36, line 60-63, 67-68 and a newly cited reference [3])
We changed in manuscript as follow “
- Line 31: In vivo animal model preclinical assessment showed that the urchin-like Ag–Au bimetallic nonwoven mat group showed significant good wound recovery because of sebaceous gland, hair follicle,
- Line 36: pure PAN nonwoven mats, was found to be a potential ideal scaffold for bone tissue engineering as osteoblast ingrowth from the top to the bottom of the membrane and proliferation inside
- Line 60-63: From the different types of solid matrix material available, electrospun polyacrylonitrile (PAN) has been widely investigated is an important engineering polymer material owing to its thermal stability, highly mechanical properties, and chemical resistance, and enhanced mechanical properties
- Line 67-68: In the musculoskeletal system, the most frequently noted effects of high-energy trauma ( missile, vehicle traffic injury , crush or blast injury, falling from heights) -energy trauma are skin and soft tissue wound defects and even loss of some bony tissue.
- “The authors did not perform any experiment expected test with growing osteoblast to clarify if the mats are sufficient for bon regeneration. It has to be done more precisely or the authors have to change the „current of the manuscript“ and avoid the hypothesis of bone regeneration without substantiation of the experiments.”
Response to Reviewer comment No. 2:
We thank this reviewer’s careful reading and constructive comments. Yes, we have made it clear that this proof-of-concept result showed that the PAN nonwoven mat is a potential ideal biomaterial for use as a bone graft material, which strikes a balance between biocompatibility, mechanical properties, osteoblast biology response, and osteoblast induction. We change our title as “Evaluation of Polyacrylonitrile Nonwoven Mats and Silver-Gold Bimetallic Nanoparticle-Decorated Nonwoven Mats for Potential Promotion of Wound Healing In Vitro and In Vivo and Bone Growth In Vitro"We also add the method “ 2.12 Culture and functional assay of osteoblasts/Immunofluorescence staining” P.6 line 236-245 in highlighted with color yellow which mention the test with growing osteoblast. This brief conclusion is based on a cell model test, as shown in Figure 9. Highly expression of the Cbfa1 (green staining), one of the major osteoblast transcription factors, indicates increase activity in these cells. Strongly labeling of stress fibers-like cytoskeleton (actin, red staining) indicates cell adhesion and osteoconductivity of PAN.
(see page 11, line 434-439, highlighted with color yellow).
- “Which type of MTT assay was used? How many experiments were done and how many repetitions?”
Response to Reviewer comment No. 3:
We thank this reviewer for his/or her detailed review. In short, extract solution method was adopted to conduct the MTT assay. We would like to discuss this for more details in materials and method section. The detailed are as follows: (P4 line149-167) NIH-3T3 cells (5 × 104/well) were seeded in 24-well plates, and medium was added to obtain a total volume of 1 mL; subsequently, the plates were cultured in an incubator for 24 h (1st well plate). Then, 5 mg of pure PAN, Ag/PAN, and Ag–Au/PAN nonwoven mats was weighed and irradiated with UV light (254 nm) for 30 min sterilization. The same concentration of commercially available nano-Ag suspension kept steady for 24 h was used as the positive control, and medium without soaking of any samples was used as the negative control. The weighed nonwoven mats were immersed in 2 mL serum-free medium in another 24-well plate (2nd well plate) and placed in the incubator for 24 h. After 24 h of extraction, the medium in the 1st well plate was removed and the cells were washed twice with 1 mL phosphate-buffered saline (PBS). Subsequently, 500 μL of the nonwoven mat’s extract solution was taken from the 2nd well plate and added to the cell plate (1st well plate). After the predetermined treatment time, the medium was removed from the plate and MTT solution (thiazol blue tetrazolium bromide [MTT; M2128-1g; 98%; Sigma-Aldrich), 0.5 mg/mL)] was added to each well to react with enzyme in the mitochondrial of the cell; the plate was subsequently incubated for 2 h. Then, MTT solution was replaced by DMSO and the plate was incubated for 10 min. Next, absorbance was measured at 570 nm using a UV-Visible spectrophotometer. We have added some more sentences to materials and method section to build strong the description (please see page 4, line 150-152, highlighted with color yellow). Each avg. value shown in Figure 4 was obtained from n = 3 repeats.
- “In generally, the authors have to complete the methods with the information about number of animals, number of repetition of experiments. There is missing the description of which type of osteoblast was used, and cultivation conditions. There is missing the description of statistical analysis, software used and type of test in all experiments.”
Response to Reviewer comment No. 4:
We thank this reviewer for the constructive comment. Per your request, we have added the statistical analyses information in the materials and method section (please see page 6, line 246-250, paragraph 2.13, highlighted with color yellow). In addition, all number of animal use or repetition of experiments were annotated in the revised manuscript (please see those color blue annotations, line152, 210 and 216). The type of osteoblast and cultivation condition have added in materials and method paragraph 2.12(please see P6 line235-45, highlighted with color yellow)
5.” It is not clear, thy author divide sensitization and irritation experiments and why used two types of animals and for wound healing the third type of animal? The description of the in vivo experiment in methods are different than description of time points on the figure. It has to be change and clarify.”
Response to Reviewer comment No. 5:
Again, we thank this reviewer for his/or her valuable comments. The decision on the animal species was based on ISO 10993-10 guideline (please see page 9 line 361 and the cited reference {20}). In the wound healing, the mice were adopted as their thickness of full skin is similar to that of human and the their sizes are convenient for animal fixation to conduct full skin removal surgery. We have added a short paragraph in the revised manuscript to benefit our broad readership. (please see page 5, line 188-191, highlighted with color yellow)
- “In the methods the authors write about the urchine like nonwoven mats, and calcined nonwoven mats are not describe. Through the results the authors sometimes analyzed calcined Ag/PAN, sometimes urchine. It is not cleare which type of the mats were analyzed in which experiment or if authors named the mats differently. It has to be clarified.
Response to Reviewer comment No. 6: We thank this reviewer for the valuable comments, and we are sorry about the unclear description of the rationale in the manuscript. Please see page 3-4, line126-131, for the detailed production procedure of the calcined AgPAN.
In manuscript as follow:
P3-4, line 126-130 “For the production of calcined Ag/PAN nonwoven mats, we placed the AgPAN into the tubular high-temperature furnace, the atmosphere inside was evacuated to 30 mmHg, and filled with nitrogen. We set the heating procedures as follows: heating to 900 ℃ with rate 5 ℃/min and the temperature was held at 900 ℃ for 1 h. After it was naturally cooled down to room temperature, the sample was taken out. The whole process was carried out at a rate of 50 mL/min under nitrogen gas flow.”
- " There is missing discussion in part of the release study, if the release of Ag, Au is sufficient in comparison with other studies.”
Response to Reviewer comment No.7: We thank this reviewer for his/or her comment. Here we would like to address this issue as follows: This release concentration issue needs to be discussed in conjunction with their antibacterial test. In Figure 5E diameter, 2.74 ± 0.33 mm and Figure 5F, 0 count of CFU after 5 h of treatment at avg., 0.062 ± 0.03 ppm of Ag ion concentration), we found that because of their improved porosity, Ag ion release efficiency and synergistical effect of Ag-Au co-release. The effective dosage of silver ion was relatively low as compared to another research works (Doaa Safwat Mohamed et al. reported that 6-12 ppm to reach the same colony reduction toward S. aureus). [add reference19, line 515, highlighted with color yellow]
Page 9, line 344-348, for newly added paragraph highlighted with color yellow:
This Ag-Au synergistical effect generate remarkable reduction in the Ag ion needed for showing antibacterial performance, which also reduce the cytotoxicity toward normal cells.
- “the clear description of figure is missing, It is not clear what is A and B. There is presented different type of graphs, why? There is visible significant toxicity, without proper discussion in the text. It is completely unclear the descriptions on line X”.
Response to Reviewer comment No. 8: We have redrawn the Figure 4 to make the format of Figure 4A and Figure 4B consistent.
- In the text, where authors describe data from Fig. 6 is written different type of mats, than is presented on the Fig. 6. Moreover the authors presented only 3 days of experiment and on the Fig. 7 the 7 days of experiment. Why?”
Response to Reviewer comment No. 9:
We appreciate this reviewer for the careful reading and review efforts. Our sensitization and irritation tests were based on ISO10993-10 guidelines, which suggest suitable animal species and time-period 3 days and 1-3 weeks to see the acute and long-term response of the sensitization and irritation evaluation. We have added this information and highlighted with color yellow in response to reviewer comment (see page 5, line 188-191, highlighted with color yellow)
In manuscript as follow:
"Our sensitization and irritation tests were thus based on ISO10993-10 guidelines, which suggest suitable animal species and time-period 3 days and 1-3 weeks to see the acute and long-term response of the sensitization and irritation evaluation."
- “ The authors presented the healing potency of prepared mats, however on left site is different numbering of days than on the right side. There is not clear which days belong to which picture?”
Response to Reviewer comment No. 10:
In Figure 8, As we conducted two runs of animal studies for evaluating two major different nonwoven mats, i.e., Ag-based/PAN and urchine like Ag-Au/PAN nonwoven mats. The left scale bar was versus group PC, NC, Ag/PAN, and calcined Ag/PAN nonwoven mats. The right scale bar was versus Ag-Au/PAN nonwoven mats.
11.“. The authors presented on Fig.9, the data which is not describe in the methods. There is missing the ruler on the Picture and there is missing the discussion about spreading, morphology and viability of the cells. On the other hand the authors describe in methods the pathology analysis, which is not describe in results and discussion.”
Response to Reviewer comment No. 11:
Thanks for your comment.
- The osteoblast seeded in PAN was added to method 2.12 Culture and functional assay of osteoblasts/Immunofluorescence staining (P.6 line 236-245 in highlighted with color yellow) which mention the test with growing osteoblast as follow:
The osteoblasts were isolated from 1-day-old new born rats as previously described[15]. Briefly, osteoblasts were harvested from calvariae and cultured in α-MEM medium with 10% FBS, 0.05% ascorbic acid (sc-394304, Santa Cruz Biotechnology, USA), 2 mMβ-glycerophosphate (G9422, Cayman Chemical, USA), 10 nM dexamethasone (sc-29059B, Santa Cruz Biotechnology, USA). Medium were changed every 3 days. 105/cm2 osteoblasts were seeded into 6-well plate containing PAN with the same medium for 28 days. Then, osteoblasts were fixed and labeled with primary anti-Core-binding factor alpha1 (Cbfa1, ab23981, Abcam USA) antibody and FITC-conjugated secondary antibody. Actin was labeled by iFluor™ 546-Phalloidin (22663, AAT Bioquest, USA). Cell Nuclei were visualized by DAPI staining. Imagines of the fluorescently stain osteoblast cells were taken by using confocal microscope (LSM 880, Zeiss, Oberkochen, Germany).
- We also revised the Figure 9 with X Y Z axis with scale. We add the discussion about the spreading, morphology, and viability of osteoblast in result and discussion. Please see P.11 line 434-441, highlighted with color in yellow as follow:
 In experiment showed in Figure 9, osteoblasts were seeded into PAN nonwoven mat and cultured for 28 days. Images of confocal microscopy showed group of osteoblasts covered on the surface as well as grew in 100 mm deep of PAN. Highly expression of the Cbfa1 (green staining), one of the major osteoblast transcription factors, indicates increase activity in these cells. Strongly labeling of stress fibers-like cytoskeleton (actin, red staining) indicates cell adhesion and osteoconductivity of PAN. Together, these results suggest osteoblasts survive and may proliferate in PAN. This proof-of-concept result showed that the PAN nonwoven mat is a potential biomaterial for use as a bone graft material, which strikes a balance between biocompatibility, mechanical properties, osteoblast biology response, and osteoblast induction.
- We revised the legend of Figure 9 , P11 line 443-447, highlighted with color in yellow as follow:
” Figure 9 Confocal imagines of osteoblasts cultured on PAN nonwoven mats: (a) Top view of 3D confocal image of osteoblast seeding into specimen. The 130 mm x 130 mm confocal image were constructed from 10 consecutive 10-mm-thick sections. (b) cross-section view of the constructed image as shown in figure (a). DAPI (blue), Cbfa1 (green), and actin (red) represent the staining method for labelling the nucleus, osteoblast-specific Cbfa1, and cytoskeleton of osteoblasts, respectively.”
We hope that the revised manuscript is now in the right format for publication in Polymers
Faithfully yours,
Chia-Chun Wu

Reviewer 2 Report
In this work polyacrylonitrile nonwoven mats and mats decorated with silver and gold nanoparticles were prepared by electrospinning. A facile, green, low-temperature protocol was developed to obtain these nonwoven materials. In vitro and in vivo experiments were performed in order to evaluate the promotion of bone ingrowth and wound healing and antibacterial effects on skin tissue. The obtained materials showed potential for use as dual-functional biomaterials for bone regeneration and infection control and composite grafts for infectious bone and soft tissue defects.
Some modifications will improve the overall quality of the manuscript. Specific points requiring attention are detailed below.
Comment 1. The authors repeat twice nonwoven mat in the Keywords section.
Comment 2. In the introduction part of the manuscript literature data concerning the preparation of PAN/Ag and/or PAN/Au fibrous materials by electrospinning are missing. There are many publications showing the electrospinning of such kind of materials with antibacterial properties. None of them has been cited.
Comment 3. In the materials part there is no unit of measurement for the Mw of PAN. Add it.
Comment 4. How did you select the electrospinning parameters for the PAN and AgNO3 solution? Why have you choose to work with voltage, 17 kV; flow rate, 7 μL/min; and working distance, 12 cm?
Comment 5. From the shown SEM micrograph with bar of 1 micrometer in Figure 1A it could be easily seen that the diameters of the PAN fibers are higher than 265.1 ± 223 nm as written in the text. Exact measurement of the fiber diameters is needed.
Comment 6. Calcination procedure should be described in the part materials and methods.
Comment 7. Figure 4 is complicated and not representative. Improve the figure by using different filling of the columns.
Comment 8. Move the cited manuscript in page 10
Maneewattanapinyo P, Banlunara W, Thammacharoen C, Ekgasit S, Kaewamatawong T (2011) An Evaluation of acute toxicity of colloidal Ag nanoparticles Journal of Veterinary Medicine and Science 73, 1417-1423
to References.
Comment 9. How do you explain the decrease in cell viability after contact with AgPAN mats? It is 81,5% on the first day, 63,33% on the 3 day and 46,33% on the 5 day.
Comments 10. In the figure text of Figure 5 there is no description what is E and F?
Comment 11. The cell morphology is hardly seen form the presented images in Figure 9. Enlarge the images and insert a scale bar.
Comment 11. Figure 9C has poor quality.
Comment 12. In the part Results and discussion a critical discussion of the obtained results with the known in the literature is missing.
The authors of the manuscript do not comment previous research published in the literature concerning preparation of PAN/Ag and PAN/Au fibrous materials by electrospinning and their potential applications. The diameters of the PAN fibres were not correctly measured. Critical discussion of the obtained results with the known in the literature is missing. I recommend that the manuscript is accepted for publication in Polymers after major revisions.
Author Response
Response to Reviewer2 Comments
Dear editor and reviewers,
Many thanks for your e-mail dated 12/29/2020. We appreciate the review of our manuscript entitled “Evaluation of Polyacrylonitrile Nonwoven Mats and Silver–Gold Bimetallic Nanoparticle–Decorated Nonwoven Mats for Potential Promotion of Wound Healing In Vitro and In Vivo and Bone Growth In Vitro”. We are grateful for your generous and helpful comments on the manuscript. The manuscript has been revised and modified according to the reviewers’ concerns point-by-point below. We have made all the changes that were suggested by the reviewers. We would like to address your comments here:
For your information at the beginning of review process:
(a)Newly added paragraphs in the revised manuscript in response to the reviewers’ comments are highlighted in yellow color.
(b)For those typo errors indicated by reviewers’ are highlighted in red.
(c)For repetition or animal use numbers are highlighted in blue.
Reviewer #2
- Comment 1. The authors repeat twice nonwoven mat in the Keywords section.
Response to Reviewer comment No.1
We thank this reviewer’s careful reading. We delete the repeat keyword of nonwoven mat. We highlighted the change in P2 line 43 with color yellow.
- Comment 2. In the introduction part of the manuscript literature data concerning the preparation of PAN/Ag and/or PAN/Au fibrous materials by electrospinning are missing. There are many publications showing the electrospinning of such kind of materials with antibacterial properties. None of them has been cited.
Response to Reviewer comment No.2
We thank this reviewer for his/or her comment. We have updated recent publication which is associated with Ag with antibacterial properties research. For some of the PAN/Ag and/or PAN/Au fibrous materials survey, we already cited 4 publications in the revised manuscript (P2 , line 63-66 , please see reference [4-7] and references cited, highlighted with color yellow )
- Comment 3. In the materials part there is no unit of measurement for the Mw of PAN. Add it.
Response to Reviewer comment No. 3
We have added the molecular weight information in the materials and method section [the Mw =150 with kDa. (Page 3, line94, highlighted with color yellow)].
- Comment 4. How did you select the electrospinning parameters for the PAN and AgNO3solution? Why have you choose to work with voltage, 17 kV; flow rate, 7 μL/min; and working distance, 12 cm?
Response to Reviewer comment No. 4
These parameters setting on the purpose of maintain at stable corn-jet mode electorspinning. We change in P.3 line 107-8 as follow:
” The parameters were set as follows: voltage, 17 kV; flow rate, 7 μL/min; and working distance, 12 cm to maintain at stable corn-jet mode electrospinning".
- Comment 5. From the shown SEM micrograph with bar of 1 micrometer in Figure 1A it could be easily seen that the diameters of the PAN fibers are higher than 4 ± 123 nm as written in the text. Exact measurement of the fiber diameters is needed
Response to Reviewer comment No.5
We thank this reviewer for the careful review. We have increased the numbers of measurement and corrected the avg. diameter in the page 7 line 259, which is highlighted with color red.
- Comment 6. Calcination procedure should be described in the part materials and methods.
Response to Reviewer comment No.6
We thank this reviewer for the constructive comment. Per your request, we have added Calcination procedure in materials and method. Please see page 3-4, line126-131, highlighted with color yellow, for the detailed production procedure of the calcined AgPAN.
In manuscript as follow:
"For the production of calcined Ag/PAN nonwoven mats, we placed the AgPAN into the tubular high-temperature furnace, the atmosphere inside was evacuated to 30 mmHg, and filled with nitrogen. We set the heating procedures as follows: heating to 900 ℃ with rate 5 ℃/min and the temperature was held at 900 ℃ for 1 h. After it was naturally cooled down to room temperature, the sample was taken out. The whole process was carried out at a rate of 50 mL/min under nitrogen gas flow.
7.Comment 7. Figure 4 is complicated and not representative. Improve the figure by using different filling of the columns.
Response to Reviewer comment No.7
Again, we thank this reviewer for the constructive comment. We have redrawn the Figure 4 to make the format of Figure 4A and Figure 4B consistent.
8.Comment 8. Move the cited manuscript in page 10 Maneewattanapinyo P, Banlunara W, Thammacharoen C, Ekgasit S, Kaewamatawong T (2011) An Evaluation of acute toxicity of colloidal Ag nanoparticles Journal of Veterinary Medicine and Science 73, 1417-1423 to References.
Response to Reviewer comment No.8
We move the cited reference and change in P.8 line 329-331 (highlighted with color yellow) as follow:
” This finding is similar to that of an in vivo study by Maneewattanapinyo et al. wherein a small shaved area of skin on guinea pigs was exposed to 50 ppm AgNP (10–20 nm) for 24 h, and the animals were observed for signs of acute toxicity. [18]"
9.Comment 9. How do you explain the decrease in cell viability after contact with AgPAN mats? It is 81,5% on the first day, 63,33% on the 3 day and 46,33% on the 5 day.
Response to Reviewer comment No.9
We thank this reviewer for the valuable comment. Yes, we have mentioned about all plausible antibacterial function mechanisms in the introduction section, which means that the Ag is going to show cytotoxicity at certain level of concentration as it has the ability to interfere the normal cell function and proliferation. In conjunction with the silver ion release profile shown in Figure 3 and in comparison to our unpublished data, we found that when the concentration of Ag is above 5-6 ppm, significant cytotoxicity was observed. This why authors can see that the cell viability was decreased when that Ah concentration is starting to increase as time pass by from 1 h , 3 h, and 5 h. We have added this description in the revised manuscript to benefit our broad readership. (please see page 8, line 321-324, highlighted with color yellow).
In manuscript as follow:
”In general, in conjunction with the silver ion release profile shown in Figure 3 and in comparison to our unpublished data, we found that when the concentration of Ag is approximately above 5-6 ppm, significant cytotoxicity was observed. This why we can see that the cell viability was decreased when the Ag concentration is starting to increase as time pass by from 1 h, 3 h, and 5 h.”
10.Comments 10. In the figure text of Figure 5 there is no description what is E and F?
Response to Reviewer comment No. 10
We thank this reviewer’s careful reading and constructive comments. We add the caption of figure 5E and E in P8 line 342 as follow” Urchin-like Ag–Au/PAN nonwoven mats showed a significantly enhanced antibacterial effect toward S. aureus (Figure 5E diameter, 2.74 ± 0.33 mm and Figure 5F, 0 count of CFU after 5 h of treatment at avg., 0.062 ± 0.03 ppm of Ag ion concentration) because of their improved porosity, Ag ion release efficiency and synergistical effect of Ag-Au co-release.” We also added the legend of Figure (e) and (f) in P9 line 357-58, highlighted with color yellow.
In manuscript as follow:
“(e) qualitatively inhibition zone results of urchin-like Ag–Au/PAN nonwoven mats and (f) quantitively sterilization rate of urchin-like Ag–Au/PAN nonwoven mats.”
11.Comment 11. The cell morphology is hardly seen form the presented images in Figure 9. Enlarge the images and insert a scale bar. Comment 11. Figure 9C has poor quality.
Response to Reviewer comment No. 11
Thanks for your comment.
We revised the Figure 9 with X Y Z axis with scale. We add the discussion about the spreading, morphology and viability of osteoblast in result and discussion P11 line 434-441, highlighted with color in yellow as follow:
”In experiment showed in Figure 9, osteoblasts were seeded into PAN nonwoven mat and cultured for 28 days. Images of confocal microscopy showed group of osteoblasts covered on the surface as well as grew in 100 um deep of PAN. Highly expression of the Cbfa1 (green staining), one of the major osteoblast transcription factors, indicates increase activity in these cells. Strongly labeling of stress fibers-like cytoskeleton (actin, red staining) indicates cell adhesion and osteoconductivity of PAN. Together, these results suggest osteoblasts survive and may proliferate in PAN. This proof-of-concept result showed that the PAN nonwoven mat is a potential ideal biomaterial for use as a bone graft material, which strikes a balance between biocompatibility, mechanical properties, osteoblast biology response, and osteoblast induction.
We also revised the legend of Figure 9, P11 line 443-447, highlighted with color in yellow as follow:
” Figure 9 Confocal imagines of osteoblasts cultured on PAN nonwoven mats: (a) Top view of 3D confocal image of osteoblast seeding into specimen. The 130 mm x 130 mm confocal image were constructed from 10 consecutive 10-mm-thick sections. (b) cross-section view of the constructed image as shown in figure (a). DAPI (blue), Cbfa1 (green), and actin (red) represent the staining method for labelling the nucleus, osteoblast-specific Cbfa1, and cytoskeleton of osteoblasts, respectively.”
- Comment 12. In the part Results and discussion a critical discussion of the obtained results with the known in the literature is missing.
Response to Reviewer comment No. 12
We thank this reviewer for the critical review. Actually, the main focus of this study is focusing on the potential application of this urchin like Ag-Au/PAN nonwoven mats for future development of a double-sided prosthesis for dual modality use, such as the treatment of open segmental fractures with soft tissue defects. However, to the best of our knowledge, we could not find any relevant literature associated with the development of double-sided prosthesis. The only way of discussion in comparison to the current literature is compared to PAN, Ag/PAN or Au/PAN only nonwoven mats. We have addressed this issue in the introduction section (please see cited reference [1-14] in page 2.
We hope that the revised manuscript is now in the right format for publication in Polymers
Faithfully yours,
Chia-Chun Wu

Round 2
Reviewer 1 Report
The authors significantly improve presentation of their results in manuscript, as well as clarified methods and complete discussion.
The manuscript is prepared in this form for acceptation.
Reviewer 2 Report
Accept the manuscript in the present form!